# Allopurinol versus Febuxostat: A New Approach for the Management of Hepatic Steatosis in Metabolic Dysfunction-Associated Steatotic Liver Disease

**DOI:** 10.3390/biomedicines11113074

**Published:** 2023-11-16

**Authors:** Amani Al-Shargi, Amal A. El Kholy, Abdulmoneim Adel, Mohamed Hassany, Sara M. Shaheen

**Affiliations:** 1Department of Clinical Pharmacy, Faculty of Pharmacy, Ain Shams University, Cairo 4393005, Egypt; amal.elkhouly@pharma.asu.edu.eg (A.A.E.K.); sara.shahin@pharma.asu.edu.eg (S.M.S.); 2National Hepatology and Tropical Medicine Research Institute (NHTMRI), Cairo 4260010, Egypt; gastrocent@yahoo.com (A.A.); mohamed.hassany@mohealth.gov.eg (M.H.)

**Keywords:** allopurinol, febuxostat, metabolic dysfunction-associated steatotic liver disease (MASLD), hepatic steatosis, xanthine oxidase inhibitor, CAP score, hyperuricemia, serum uric acid (SUA), non-alcoholic fatty liver disease (NAFLD)

## Abstract

Metabolic dysfunction-associated steatotic liver disease (MASLD) includes patients with hepatic steatosis and at least one of five cardiometabolic risk factors. Xanthine oxidase (XO) represents a treatment target for MASLD. We aimed to evaluate the effect of two xanthine oxidase inhibitors, allopurinol and febuxostat, plus lifestyle modifications compared to lifestyle modifications alone on improving steatosis. Ninety MASLD patients were assigned to one of three groups for three months. Patients with hyperuricemia were given either allopurinol 100 mg or febuxostat 40 mg daily, along with lifestyle modifications. The third control group was only given lifestyle modifications, excluding all patients with hyperuricemia due to ethical concerns. The primary outcome was to measure the change in the controlled attenuation parameter (CAP) score as an indicator of steatosis from baseline after three months. The secondary outcome was to measure the change in serum uric acid (SUA) three months from baseline. The study found that the CAP score decreased significantly in the allopurinol group (*p* = 0.009), but the decline in the febuxostat or lifestyle groups was non-significant (*p* = 0.189 and 0.054, respectively). The SUA levels were significantly reduced in both the allopurinol and febuxostat groups (*p* < 0.001), with no statistical difference between the two groups (*p* = 0.496).

## 1. Introduction

Metabolic dysfunction-associated steatotic liver disease (MASLD), previously known as non-alcoholic fatty liver disease (NAFLD), is a multisystemic disease characterized by hepatic steatosis and associated cardiometabolic disorders, such as obesity, insulin resistance (IR), dyslipidemia, and hypertension. It is considered one of the leading causes of liver morbidity and mortality worldwide. It affects one-quarter of the world’s adult population and is the second major cause of advanced liver disease and liver transplants in Europe and the United States [1]. MASLD can progress from simple steatosis to metabolic dysfunction-associated steatohepatitis (MASH), which is characterized by inflammatory infiltration and can lead to hepatic fibrosis, cirrhosis, and hepatic carcinoma [2].

Estimating liver fat (steatosis) plays a crucial role in determining MASLD. A liver biopsy (LB) is the gold standard for steatosis severity evaluation and diagnosis. However, it is an invasive approach that is difficult to replicate, and most patients are unprepared for it [3,4]. The CAP value measured by FibroScan^®^ (Echosens, Paris, France) provides a non-invasive alternative to LB for detecting and measuring steatosis (S) with reasonable accuracy, lower cost, and without any implications. CAP reflects a decrease in the amplitude of the ultrasound signal in the liver and is measured in decibels per meter (dB/m) [5]. Figure 1 provides a schematic view of the differences between FibroScan^®^ and a liver biopsy.

Hyperuricemia is a risk factor for MASLD, and it is significantly higher in patients with MASLD than controls [6,7]. The SUA level used to identify hyperuricemia differs slightly between studies; however, the most commonly used threshold is 6 mg/dL, which is the level at which monosodium urate precipitates in the joints [8]. In MASLD, hyperuricemia is assumed to be mediated by increased hepatic XO expression and activity, which catalyzes the oxidation of hypoxanthine to xanthine and then to uric acid (UA) [9]. UA has been established to promote de novo lipogenesis and cause insulin resistance (IR) in vivo and in vitro by enhancing the formation of reactive oxygen species (ROSs), superoxide anions, and hydrogen peroxide [10]. In addition to IR, it is believed that higher levels of inflammatory cytokines, ROSs, and subsequent lipid peroxidation contribute to the progression of MASLD toward liver cirrhosis and hepatocellular carcinoma [11].

Allopurinol and febuxostat are xanthine oxidase inhibitors often prescribed to treat gout. Studies have shown promising results in using these drugs to treat MASLD. For example, in mice with high-fat diet-induced obesity, allopurinol was found to relieve insulin resistance and hepatic steatosis by reducing hepatic XO levels and activity [12]. In another study, febuxostat reduced hepatic XO activity effectively and UA levels in MASH model mice [13]. Therefore, we investigated the effects of pharmacologically suppressing XO in MASLD patients with hyperuricemia using allopurinol or febuxostat.

## 2. Materials and Methods

### 2.1. Study Design

The study was a prospective, interventional, open-label, cluster-randomized clinical trial conducted on adult patients at the Liver Unit of the National Hepatology and Tropical Medicine Research Institute, Cairo, Egypt, from January 2022 to January 2023.

### 2.2. Patient Eligibility (Inclusion and Exclusion Criteria)

All patients were evaluated for eligibility and enrolled in the study if they met the following inclusion criteria: male or female between the ages of 18 and 65; presence of hepatic steatosis confirmed by previous ultrasound imaging, along with CAP scores of more than 238 dB/m; SUA levels greater than 6 mg/dL (in allopurinol and febuxostat groups); and diagnosis of cardiometabolic syndrome according to the National Cholesterol Education Program Adult Treatment Panel (NCEP ATP III) definition and the International Diabetes Federation (IDF) [14]. Patients with SUA levels greater than 6 mg/d who required urate-lowering treatments were excluded from the control group due to ethical concerns.

The exclusion criteria for this study involved certain medical conditions, including renal insufficiency (indicated by serum creatinine levels over 2.0 mg/dL), complex coronary heart disease, heart failure (cardiac function grade 2 or above), asthma and other pulmonary disorders, a previous episode of viral hepatitis or serological testing indicating viral hepatitis, and secondary causes of fatty liver disease, like autoimmune disease. Other exclusion criteria included alcohol consumption, use of uric-lowering medications in the four weeks preceding screening, medications that may induce fatty liver, such as tamoxifen and steroids, a history of allergy to febuxostat and allopurinol, pregnancy, or breastfeeding.

### 2.3. Enrollment and Allocation

A total of 200 patients were assessed for eligibility, and 110 patients were excluded (96 patients did not meet the inclusion criteria, and 16 patients refused to participate in the study). Ninety MASLD patients were distributed into two clusters. The first cluster consisted of 60 patients with hyperuricemia who received treatment, while the second cluster consisted of 30 patients with hyperuricemia who received lifestyle modifications. The duration of the study was three months. The 60 patients with hyperuricemia were randomly divided using a random block method into 2 groups, with 30 patients receiving allopurinol with lifestyle modifications and the remaining 30 receiving febuxostat with lifestyle modifications. None of the participants had received lifestyle interventions before the trial. Figure 2 represents the process.

### 2.4. Treatment Intervention

In this study, MASLD patients were separated into three groups. Patients with MASLD who had hyperuricemia were given allopurinol or febuxostat along with lifestyle modifications. A total of 30 patients were administered allopurinol 100 mg/day, made by GlaxoSmithKline under the trade name Zyloric^®^, and were called the allopurinol group. In comparison, the other 30 patients were given febuxostat 40 mg/day, manufactured by Hikma Pharma S.A.E.-Egypt under the trade name Feburic^®^, and were called the febuxostat group. The third group was the control group, which included 30 MASLD patients without hyperuricemia who just received lifestyle treatments and were called the lifestyle group.

The lifestyle modification included a healthy diet, exercise, and gradual weight loss, which are recommended in all guidelines for treating MASLD patients. (Asia-Pacific guidelines, EASL, NICE, and AASLD guidelines) [15,16]. Scientific research suggests that a 3–5% reduction in body weight is sufficient for improving steatosis and fibrosis in MASLD patients, and this percentage served as the baseline for weight loss in our study [17].

### 2.5. Clinical and Laboratory Data

Upon entering the study, patients underwent a comprehensive medical history and physical assessment, which included measuring their body mass index (BMI) and waist circumference (WC). Additionally, baseline biochemical laboratory tests were conducted, including a complete blood count (CBC), SUA, alanine transaminase (ALT), aspartate transaminase (AST), serum creatinine, glucose, and lipid profile.

SUA was measured using the enzymatic colorimetric method (uricase/PAP test), and serum creatinine was determined using the photometric method (Pars biochemical kits). Regarding the lipid profile, including LDL-C, HDL-C, and TG, they were determined using enzymatic colorimetric tests manufactured by Human Gesellschaft fur Biochemica und Diagnostica mbH-Germany. ALT and AST were determined by using the International Federation of Clinical Chemistry (IFCC) kinetic methods [18].

All participants underwent ultrasounds using My Lab (EsaoteTM Class C with a linear probe 7/5–12 MHz, Germany (Esaote Biomedica Deutschland GmbH, Köln, Germany). A Fibroscan of the liver was performed by FibroScan^®^ 502 Touch (SNF01686) equipped with both M and XL probes (SN90649) (A 2.2.0.0). After that, the CAP was determined, and participants with CAP values over 238 dB/m were included. The conventional cut-off point for the CAP score was used as follows: S1 for steatosis between 238 and 259 dB/m; S2 for scores between 260 dB/m and 291 dB/m; and S3 was described as a CAP score of 292 dB/m or higher [19].

Experienced and accredited clinicians utilized FibroScan^®^ to measure CAP through transient elastography. The device software incorporated a programmed probe selection tool to choose the appropriate probes. In cases where the M probe failed, the XL probe for obese patients was used instead [20]. Patients were positioned supine with their right arm fully abducted, and measurements were taken by imaging the right liver lobe via an intercostal space. CAP is an average prediction for ultrasonic attenuation at 35 MHz, presented as decibels per meter. Only measures with a minimum of ten valid measurements per participant were considered acceptable [21].

### 2.6. Follow Up

Patients were instructed to engage in moderate-intensity physical activity for at least 30 min and more than twice per week. They were also advised to follow a healthy diet of low-to-moderate fat and carbohydrate intake and a calorie restriction of 500 to 1000 kcal per day to achieve gradual weight loss [15,16]. Participants were followed up monthly at the clinic for medication refills and were educated regarding medication side effects and medication adherence. After three months, patients’ weight, BMI, WC, and laboratory markers, such as serum uric acid, AST, ALT, serum creatinine, and steatosis, were evaluated.

### 2.7. Safety and Tolerability

Liver enzymes were monitored for hepatotoxicity during the three-month course of the medication. “Hepatotoxicity” is defined as AST/ALT elevation to at least three times the upper normal range when the initial value of AST/ALT was regular or twice the initial value of AST/ALT when the initial value of AST/ALT was elevated [22]. Throughout the trial, patients were observed for any treatment-related adverse effects, such as medication intolerance, diarrhea, headaches, joint-related signs and symptoms, connective tissue signs and symptoms, and musculoskeletal issues.

### 2.8. Sample Size Calculations

We predicted that a sample of 75 patients (25 per group) would be needed to detect a difference in the CAP score of 28.2 dB/m (9% from baseline) with 80% power at a significance level of 0.05 [19]. Assuming up to a 20% dropout rate, the sample size was 90 participants (30 per group).

### 2.9. Statistical Analysis

IBM SPSS^®^ Statistics version 26 (IBM^®^ Corp., Armonk, NY, USA) was utilized for statistical analysis. Numerical data were presented as mean, standard deviation, median, and range. Frequency and percentage were used to express qualitative data. The association among qualitative variables was investigated using Pearson’s chi-square test or Fisher’s exact test. Quantitative data were evaluated for normality using Shapiro–Wilk tests. The comparison between the three groups was made using either analysis of variance (ANOVA) for normally distributed numeric data or the Kruskal–Wallis test (non-parametric ANOVA) for not normally distributed data. Then, the “post-hoc test” was applied for pair-wise comparison.

The comparison between two consecutive measures of numerical variables was performed using either a paired *t*-test or the Wilcoxon signed rank test (non-parametric paired *t*-test). The *p*-value was adjusted using the Bonferroni procedure due to multiple comparisons. The correlation between numerical variables (change in BMI, change in CAP score, and change in SUA) was assessed using Spearman’s correlation. All tests were two-tailed, with a *p*-value < 0.05 considered significant.

## 3. Results

Eighty-three MASLD patients completed the study and were analyzed; twenty-eight people were in the allopurinol group, twenty-six in the febuxostat group, and twenty-nine in the lifestyle group. The patient enrollment flow diagram and the reasons for dropout are shown (Figure 2).

### 3.1. Characteristics of Study Participants

The study participants were evaluated based on their demographic, laboratory parameters, and CAP score at baseline, and there were no significant differences among the three study groups, except for SUA, since the patients in the control group did not have hyperuricemia (Table 1). Therefore, we compared the allopurinol and febuxostat groups in terms of serum uric acid levels before and after treatment, and there was no significant difference between the two groups at baseline (*p* = 0.260). Also, there was no significant difference between the two groups in terms of serum uric acid level and SUA percent change at the end of the study (*p* = 0.496 and *p* = 0.993, respectively). Here, the percent change refers to 100 * (value after—value before)/value before.

In addition, Table 1 demonstrates a comparison within each group before and after treatments in terms of laboratory parameters. Both the allopurinol and febuxostat groups showed a significant decrease in SUA levels (*p* < 0.001). Regarding serum creatinine and liver enzymes (ALT and AST), there were no significant differences among the three groups at the end of the study, and there were no significant changes within each group.

The data represented in Table 1 show that at the end of the study, weight, BMI, and WC all decreased with no statistically significant difference between the three groups (*p* > 0.05). However, there was a significant decrease within each group in terms of weight, BMI, and WC (*p* < 0.05).

### 3.2. Liver Steatosis (CAP Score)

As shown in Figure 3, the allopurinol group had a substantial decrease in CAP score (*p* = 0.009), dropping from 342 (267–400) dB/m before treatment to 304.5 (233–400) dB/m after treatment. The CAP score in the febuxostat group decreased from 311 (240–400) dB/m before treatments to 287.5 (221–396) dB/m after treatments; however, this decrease was not statistically significant (*p* = 0.189). The lifestyle group also reduced the CAP score from 318 (239–400) dB/m to 300 (115–357) dB/m, but this was not statistically significant (*p* = 0.054).

At the end of the study, there was no significant difference between the three groups based on the CAP score (*p* = 0.651). The allopurinol group had the most significant percentage decrease (−9.48 ± 14.04), followed by the lifestyle group (−9.23 ± 18.59), and, finally, the febuxostat group (−5.21 ± 13.79). However, there was no significant difference between the three groups (*p* = 0.486). The percentage of patients who improved their CAP score was 82.1% in the allopurinol group, 79.3% in the lifestyle group, and 76.9% in the febuxostat group, with no significant difference between the three groups (*p* = 0.893).

Table 2 demonstrates that the percentage of patients with significant steatosis (S3 ≥ 292 dB/m) decreased from 89.3% to 57.1% in the allopurinol group, from 72.4% to 51.7% in the lifestyle group, and from 53.8% to 42.3% in the febuxostat group. However, there was no significant difference between the three groups (*p* = 0.680).

The conventional cut points of the CAP scores are as follows: S1 ≥ (238 dB/m), S2 ≥ (259 dB/m), and S3 ≥ (292 dB/m). The *p*-value is for the difference between the three groups: *p*-value before = 0.056 (Pearson’s chi-square), and *p*-value after = 0.680 (Fisher’s exact test).

The study found that there was a significant positive correlation between the change in CAP score and the change in BMI for all participants (r = 0.243, *p* = 0.027), and the febuxostat group had a strong positive correlation between the two variables (r = 0.455, *p* = 0.019) (Table 3). Conversely, we did not find any significant correlation between the percent change in UA and the percent change in CAP score in the whole group of all participants (*p* = 0.284).

### 3.3. Adverse Events

During the study, participants experienced mild adverse events such as diarrhea, headaches, joint-related signs and symptoms, and musculoskeletal and connective tissue signs and symptoms. There were no serious incidences of rashes or hypersensitivity reactions. The incidence of acute gout flares increased during the early stage of urate-lowering treatment. When seven patients in the allopurinol group and nine patients in the febuxostat group experienced an acute gout flare, they were prescribed ibuprofen 800 mg three times daily within the first 24 h of onset to reduce the intensity and duration of the attacks. Treatment with ibuprofen lasted 5 to 7 days until the attack was cured [23]. None of the patients discontinued allopurinol or febuxostat medication during the gout flare. The gout flare was regarded as a typical gout symptom rather than an adverse event.

### 3.4. Hepatotoxicity

Allopurinol and febuxostat had no hepatotoxicity since there was no AST/ALT elevation to three times the upper standard limit when the baseline AST/ALT was normal or no duplication in the initial AST/ALT when the initial AST/ALT was high.

## 4. Discussion

This study aimed to investigate the effects of febuxostat 40 mg and allopurinol 100 mg, along with lifestyle modifications, on hepatic steatosis improvement in MASLD patients. Previous studies demonstrated that the CAP score is a non-invasive, low-cost approach for diagnosing and quantifying steatosis [5].

Our study showed that the CAP score significantly decreased in the allopurinol group, but the decline in the other two groups was non-significant. In addition, there was a substantial reduction in serum uric acid in both the allopurinol and febuxostat groups, with no significant difference between the two groups.

In agreement with our results, Wan et al. [12] observed that uric acid promotes hepatocyte fat accumulation, insulin resistance, and insulin signaling disruption both in vivo and in vitro. They also observed that uric acid enhanced NOD-like receptor family pyrin domain containing 3 (NLRP3) inflammasome activation, but allopurinol decreased NLRP3 inflammasome activation in a high-fat-diet mouse model of NAFLD. Furthermore, knocking down NLRP3 expression dramatically reduced uric acid-induced fat formation within HepG2 and L02 cells. Knocking down NLRP3 expression also restored uric acid-induced insulin signaling dysfunction in both cell types.

Furthermore, Xu et al. [24] found that XO expression and activity increased in a free fatty acid-induced cellular model of MASLD, while XO gene knockdown significantly inhibits uric acid and ROS production, leading to reduced hepatocyte fat accumulation in the cellular model of MASLD, as well as decreasing NLRP3 inflammasome activation in free fatty acid-exposed HepG2 cells. The production of redox species or positive divalent cations can negatively affect cellular [25] or biomolecular [26] levels, and they could trigger the progression of human hepatic diseases.

In a previous study [27], febuxostat was found to be effective in preventing the development of MASH in mice. Febuxostat normalized hyperuricemia, elevated ALT, and increased tunnel-positive cells in the liver in mice fed a high-fat diet. In another study, febuxostat and allopurinol decreased the progression of hepatic steatosis, fibrosis, and glucose intolerance in MASH model mice [28]. They also conducted human pilot research with febuxostat in MASLD patients with hyperuricemia to see if it lowers serum levels of ALT and AST, two signs of liver impairment. They discovered that febuxostat relieved MASLD in hyperuricemia patients, according to changes in the serum levels of uric acid, ALT, AST, and lactate dehydrogenase, resulting in decreased IR, peroxidation of lipids, and classically activated M1-like buildup in the liver. On the other hand, our study did not find any statistically significant difference between the three groups in terms of liver enzymes (AST, ALT) and serum creatinine at the end of the study.

According to the correlation, our study found a significant positive correlation between changes in CAP and BMI in the whole group of participants, confirming the beneficial effect of lifestyle changes such as diet and exercise on the outcomes. The results are consistent with Hallsworth and Adams [17], who demonstrated that weight loss from a healthy lifestyle improves steatosis, liver damage, and fibrosis in MASH patients. In contrast, the study by Shimizu et al. [29] did not find any significant correlation between the changes in CAP score and changes in body weight.

On the other side, our study did not find a significant correlation between uric acid percent change and CAP score percent change in the whole group (*p* = 0.284). However, the study conducted by Feng et al. [30] demonstrated a positive correlation between serum uric acid and the CAP score as a marker of hepatic steatosis.

Our study findings suggest that both allopurinol and febuxostat are safe for the liver and well tolerated. Neither of them caused any elevation of AST/ALT to three times the upper normal range when the baseline AST/ALT was regular, nor did they cause any doubling of the baseline AST/ALT when the baseline AST/ALT was raised. According to the National Institutes of Health and the Liver Toxic Drug Record, the incidence of abnormal liver function tests (LFTs) for allopurinol and febuxostat has been reported at 2–6% and 2–13%, respectively [31].

The study by Lee et al. [22] found that febuxostat (3/32, 9.4%) is less likely to cause hepatotoxicity than allopurinol (36/102, 35.3%, *p* = 0.005) in patients with FLD. Instead, diabetes (*p* = 0.009) and colchicine usage (*p* < 0.001) may raise the risk of hepatotoxicity.

Although this was the first study comparing the effects of allopurinol and febuxostat along with lifestyle modifications in patients with MASLD and hyperuricemia, our study has some limitations. First, a longer duration of treatment might be necessary to confirm that the decreases in CAP scores in the lifestyle and febuxostat groups were significant. Second, a larger sample size might be required to ascertain the effects. Lastly, the study did not perform a liver biopsy, the gold standard method to evaluate the status of MASLD.

## 5. Conclusions

Our study found that allopurinol 100 mg/day for three months, when combined with a healthy lifestyle, significantly lowers the CAP score, an indicator of hepatic steatosis. Also, it was found that allopurinol 100 mg/day is equally effective as febuxostat 40 mg/day in lowering blood uric acid levels. Finally, allopurinol 100 mg/day and febuxostat 40 mg/day were shown to be safe for the liver. As a result, xanthine oxidase inhibitors may be a useful treatment option for MASLD patients with hyperuricemia.

## Figures and Tables

**Figure 1 biomedicines-11-03074-f001:**
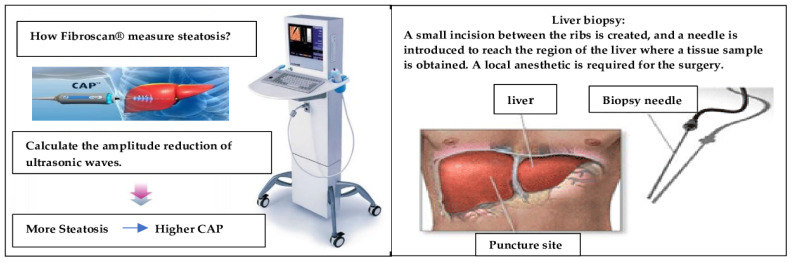
A schematic view of the differences between FibroScan^®^ and liver biopsy.

**Figure 2 biomedicines-11-03074-f002:**
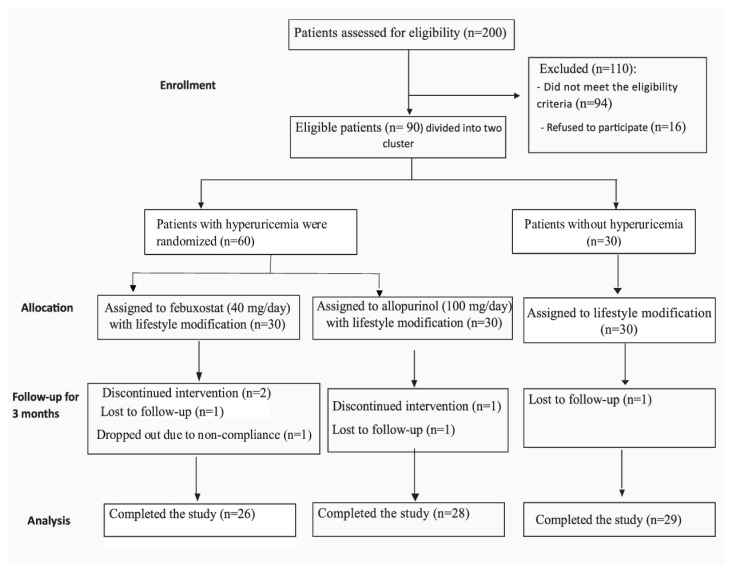
Flow diagram representing enrollment, allocation, follow-up, and analysis processes; UA denotes serum urate concentration.

**Figure 3 biomedicines-11-03074-f003:**
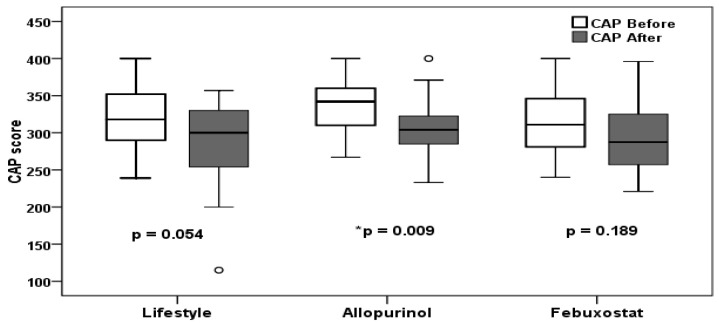
Distribution of CAP scores stratified by study groups; data are median (range); * (*p* < 0.05) with significant results when compared to baseline.

**Table 1 biomedicines-11-03074-t001:** Patient’s characteristics before and after the study.

Variable	Lifestyle, (N = 29)			Allopurinol, 100 mg/Day (N = 28)			Febuxostat, 40 mg/Day (N = 26)			*p*-Value	
	Before	After	*p*-Value *	Before	After	*p*-Value *	Before	After	*p*-Value *	Before	After
Age—years.	46.0 ± 10.3			48.1 ± 11.0			47.3 ± 11.6			0.764 ^a^	
Female sex (%)	(89.7)			(67.9)			(76.9)			0.133 ^b^	
Weight (kg)	94.2 ± 10.3	92.75 ± 15.34	**0.029 ^d^**	98.0 ± 18.6	96.0 ± 18.1	**0.001 ^d^**	100.7 ± 16.5	98.69 ± 14.95	**0.039 ^d^**	0.356 ^a^	0.401 ^a^
BMI (kg/m^2^)	36.3 ± 5.1	35.8 ± 5.3	**0.033 ^d^**	36.1 ± 6.6	35.4 ± 6.5	**0.001 ^d^**	37.8 ± 6.2	37.1 ± 5.8	**0.044 ^d^**	0.538 ^a^	0.554 ^a^
WC (cm)	104.5 ± 9.1	103.6 ± 8.7	**0.034 ^d^**	106.2 ± 10.2	105.6 ± 09.2	**0.038 ^d^**	105.7 ± 9.7	104.7 ± 8.9	**0.032 ^d^**	0.267 ^a^	0.397 ^a^
SCR (mg/dL)	0.83 ± 0.17	0.81 ± 0.13	0.387 ^d^	0.89 ± 0.23	0.90 ± 0.14	0.789 ^d^	0.87 ± 0.22	0.91 ± 0.18	0.289 ^d^	0.541 ^a^	0.252 ^a^
AST (u/L)	28.1 ± 11.2	25.2 ± 9.8	0.056 ^k^	31.8 ± 15.9	30.7 ± 10.3	0.289 ^k^	27.3 ± 11.1	30.7 ± 12.1	0.162 ^k^	0.557	0.459 ^c^
ALT (u/L)	27.9 ± 10.6	26.5 ± 14.0	0.153 ^k^	38.6 ± 25.6	33.3 ± 16.1	0.858 ^k^	28.7 ± 13.3	26.0 ± 11.0	0.545 ^k^	0.274	0.494 ^c^
AST/ALT	1.05 ± 0.33	1.05 ± 0.40	0.452 ^k^	0.93 ± 0.32	0.98 ± 0.26	0.135 ^k^	1.03 ± 0.38	1.22 ± 0.28	0.234 ^k^	0.289	0.099 ^c^
SUA (mg/dL)	3.87 ± 0.59	4.1 ± 0.90	0.090 ^d^	6.70 ± 0.90	4.1 ± 1.10	**<0.001 ^d^**	6.40 ± 0.80	3.9 ± 0.80	**<0.001 ^d^**	**<0.001 ^a^**	0.711 ^a^
HbAIc %	6.4 ± 1.2			6.6 ± 1.4			6.3 ± 1.02			0.253 ^a^	
SBP	119.1 ± 22.6			112.3 ± 11.3			113.7 ± 14.8			0.287 ^a^	
DBP	84.2 ± 4.4			85.4 ± 3.5			85.3 ± 4.1			0.487 ^a^	
TC (mg/dL)	186.9 ± 29.8			222.9 ± 63.0			199.7 ± 64.2			0.065 ^c^	
TG (mg/dL)	180.1 ± 99.4			214.0 ± 105.3			171.1 ± 63.3			0.146 ^c^	
HDL (mg/dL)	43.3 ± 8.4			42.9 ± 12.5			46.7 ± 14.2			0.417 ^c^	
LDL (mg/dL)	111.0 ± 32.0			115.2 ± 21.9			109.5 ± 32.3			0.221 ^c^	

Data are the mean ± SD for normally distributed parameters or no. (%) the *p*-value for the difference between three groups at baseline; *p*-value * for the difference within each group; ^a^ according to the ANOVA; ^b^ according to the Pearson’s chi-squared test; ^c^ according to the Kruskal–Wallis test; ^d^ according to the paired *t*-test; ^k^ according to the Wilcoxon signed rank test; BMI: body mass index; WC: waist circumference; SCR: serum creatinine; AST: aspartate aminotransferase; ALT: alanine aminotransferase; SUA: serum uric acid; HbAIc %: hemoglobin A1C; SBP: systolic blood pressure; DBP: diastolic blood pressure; TC: total cholesterol; TG: triglyceride; HDL: high-density lipoprotein; LDL: low-density lipoprotein. The statistically significant *p* values (*p* < 0.05) are in bold.

**Table 2 biomedicines-11-03074-t002:** Comparison of the CAP scores (S1, S2, and S3) between the three groups before and after the treatment.

CAP Scores		Lifestyle		Allopurinol, 100 mg/Day		Febuxostat, 40 mg/Day	
		Before	After	Before	After	Before	After
S0	Patient count	0	6	0	2	0	5
	% within the group	0.0%	20.7%	0.0%	7.1%	0.0%	19.2%
S1	Patient count	3	2	0	1	4	2
	% within the group	10.3%	6.9%	0.0%	3.6%	15.4%	7.7%
S2	Patient count:	5	6	3	9	8	8
	% within the group	17.2%	20.7%	10.7%	32.1%	30.8%	30.8%
S3	Patient count:	21	15	25	16	14	11
	% within the group	72.4%	51.7%	89.3%	57.1%	53.8%	42.3%

**Table 3 biomedicines-11-03074-t003:** Change (after–before) in BMI and CAP score in each group and the whole group.

	Lifestyle (N = 29)	Allopurinol (N = 28)	Febuxostat (N = 26)	Whole Group (N = 83)
BMI change	−0.78 (−3.81–2.22)	−0.73 (−2.94–1.56)	−0.75 (−4.57–5.27)	−0.76 (−4.57–5.27)
CAP change	−24.0 (−139.0–87.0)	−38.0 (−113.0–60.0)	−23.0 (−104.0–73.0)	−24.00 (−139.00–87.00)
Correlations in each group between CAP change and BMI change.				
(r)	0.111	0.253	0.455	0.243
*p*-value	0.568	0.194	**0.019**	**0.027**

BMI changes and CAP changes are in the median (range), *p*-value for correlation within each group according to Spearman’s rho test, N for patients’ number, and (r) for the correlation coefficient, the statistically significant *p* values (*p* < 0.05) are in bold.

## Data Availability

The original contributions presented in the study are included in the article. Further inquiries can be directed to the corresponding authors.

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
