# Peer review of "Allopurinol versus Febuxostat: A New Approach for the Management of Hepatic Steatosis in Metabolic Dysfunction-Associated Steatotic Liver Disease"

_biomedicines, 2023, doi:10.3390/biomedicines11113074_

Round 1
Reviewer 1 Report
Comments and Suggestions for Authors
The manuscript titled “Allopurinol versus Febuxostat: A new approach for management of Hepatic Steatosis in Metabolic Dysfunction-Associated Steatotic Liver Disease” by Al-Shargi, A.; et al. is a comparative study where the authors assess the effect of two different chemical compounds to treat 200 patients with MASLD for 3 months. The authors found that these compounds decrease the uric acid levels and improve CAP score levels associated to liver diseases by inhibiting the action of the xanthine oxidase enzyme.
The scientific content is interesting and the sections are well-designed. However, it exists some points that need to be addressed (please, see them below detailed point-by-point). The most relevant outcomes found by the authors can contribute to in the growth of many fields like the clinical&healthcare overall related to the treatment of liver diseases. This knowledge could aid in the design of the next-generation of therapies against liver disorders. For this reason, I will recommend the present scientific manuscript for further publication in Biomedicines once all the below described suggestions will be properly fixed.
Here, there exists some points that must be covered in order to improve the scientific quality of the manuscript paper:
1) ABSTRACT (OPTIONAL). “The secondary outcome was to measure the change in serum uric acid (…)” (line 19). The authors should consider to add the abbreviation of the terms “uric acid” and “controlled attenuation parameter” between brackets in the same way as the previously defined “Metabolic Dysfunction-Associated Steatotic Liver Disease”.
2) KEYWORDS (OPTIONAL). The authors should consider to add the term “non-alcoholic fatty liver disease (NAFLD)” in the keyword list.
3) INTRODUCTION. “It is considered one of the primary causes of liver morbidity and mortality worldwide” (lines 32-33). The authors should provide more quantitative insights about the global burden of liver diseases [1] to keep the attention to the potential readers about the importance of this topic.
[1] Devarbhavi, H.; Asrani, S.K.; Arab, J.P.; Nartey, Y.A.; Pose, E.; Kamath, P.S. Global burden of liver disease: 2023 update. J. Hepatol. 2023, 79, 516-537. https://doi.org/10.1016/j.jhep.2023.03.017.
4) “The controlled attenuation parameter (CAP) value (…) noninvasive alternative to liver biopsy (…) CAP reflects a decrease in the amplitude of the ultrasound signal in the liver and is measured in decibels per meter” (lines 40-44). The authors should add a schematic representation to better visualize the existing differences among both approaches.
5) MATERIALS & METHODS. “Liver enzymes were followed up for monitoring of hepatotoxicity” (line 147). Please, the authors should specify the time range of the indicated monitoring study.
6) RESULTS. “Adverse effects” (lines 250-258). The authors did not depict some potential side effects related to the dose of allopurinol or febuxostat drugs (body aches, blurred vision, dry eyes, ….) different to the cited in the main manuscript body text (gout flare symptoms). Did the patients showed some of these non-desirable effects? A brief discussion should be added in this regard.
7) DISCUSSION. “Furthermore, (…) XO expression and activity increased in a free fatty acid-induced cellular model of MASLD, and that XO gene knockdown significantly inhibits uric acid production and reduces hepatocyte fat accumulation (…)” (lines 277-280). Even if I agree with this information, the authors should strengthen the message that the production of redox species or positive divalent cations can negatively affect at cellular [2] or biomolecular [3] levels and they could trigger the progression of human hepatic diseases.
[2] Allameh, A.; Niayesh-Mehr, R.; Aliarab, A.; Sebastiani, G.; Pantopoulos, K. Oxidative Stress in Liver Pathophysiology and Disease. Antioxidants 2023, 12, 1653. https://doi.org/10.3390/antiox12091653.
[3] Vega, S.; Neira, J.L.; Marcuello C.; Lostao, A.; Abian, O.; Velazquez-Campoy, A. NS3 protease from hepatitis C virus: biophysical studies on an intrinsically disordered protein domain. Int. J. Mol. Sci. 2013, 14, 13282-13306. https://doi.org/10.3390/ijms140713282.
8) This section is clear and concise. The most relevant outcomes found in this work are highlighted in this section. No actions are requested from the authors.
Comments on the Quality of English LanguageThe authors should recheck the English out before to send the manuscript revised version in order to fix some aspects susceptible to be improved.
Reviewer 2 Report
Comments and Suggestions for Authors
The authors examined the impact of XO inhibitors allopurinol and febuxostat in combination with lifestyle modification as well as of lifestyle modification alone on the CAP score and uric acid serum levels. The study shows that both XO inhibitors significantly decreased uric acid but only allopurinol + lifestyle modification significantly decreased the CAP score after 3 months.
The study is well designed and conducted but data presentation is poor:
1. Results shown in Table 1,2,3 and 4 should be combined in 1 Table which should be organized as Table 3, i.e. should contain before after data. This new Table should not contain the CAP score as it is shown in Fig. 2.
All variable in the new Table should have units; this was not the case in old Table 3.
The lower part of Table 4 = % of improved CAP score in each group should be shown (described) only in the text.
Is the CAP score in Fig. 1 median and range? In Table 4 it was mean (SD) and in Table 3 median and range! The Fig 2 legend should indicate what the boxes represent.
The combined Table 1 should be systematically described- first it should be highlighted that except uric acid all baseline parameters were not significantly different between the groups (The previous Table 1 can be shown in suppl as Table S1). Then, the authors should described how the treatment affected all parameters (before vs. after) in the groups.
Here, it is very important to examine the differences in BMI, weight, and WC to show the impact of lifestyle modification on these parameters in each group. This is the only possibility to examine the efficacy of lifestyle modification.
After that the authors should focus on the CAP score and describe Fig. 2, % improvement in the CAP score (only text), CAP% change (only text) and the data presented in the new Table 2 (old Table 5).
2. Table 5 should be new Table 2.
3. Table 6 should be omitted it is repetition; CAP % change should be described in the text only.
4. Table 7 should be new Table 3.
How the authors monitored the efficacy/compliance regarding lifestyle modification?
Some questionary might have been helpful. Were there any advices given to the patients regarding healthy diet, the type and extent of physical activity?
Lane 53: please remove ( ) around IR; it was defined in the previous sentence.
Lane 127,128: controlled attenuation parameter (CAP) should be only CAP as it was explained in the Introduction.
Lane 244: SUA was not explained uric acid was UA in the Introduction section.
Lanes 272-275: Not clearly explained.
Lanes: 280-283: not clear; …were normalized in mice…BY WHAT ? Probably febuxostat but it is not clear that it is related to the previous sentence.
Lane 302: P=0.284 instead of higher than 0.05.
Lanes: 317-319: Not clear…please re-writte.
Comments on the Quality of English Language
Moderate editing required.
Round 2
Reviewer 2 Report
Comments and Suggestions for Authors
The authors followed the reviewer's suggestions and substantially improved the mansucript.
Comments on the Quality of English LanguageMinor editing needed.
Author Response
Dear Editor and Reviewers,
Kindly note that the English writing was checked as required. Please see the attachments.
Best regards,
Amani Al-shargi
